# Gendered economic, social and health effects of the COVID-19 pandemic and mitigation policies in Kenya: evidence from a prospective cohort survey in Nairobi informal settlements

Jessie Pinchoff [1], Karen Austrian [2], Nandita Rajshekhar,[3] Timothy Abuya [4], Beth Kangwana,[2] Rhoune Ochako,[2] James Benjamin Tidwell [5], Daniel Mwanga,[4] Eva Muluve,[2] Faith Mbushi,[2] Mercy Nzioki,[2] Thoai D Ngo[1]

[1]Poverty, Gender and Youth, Population Council, New York City, New York, USA
[2]Poverty, Gender and Youth, Population Council, Nairobi, Kenya
[3]Independent Consultant, New York City, New York, USA
[4]Reproductive Health, Population Council, Nairobi, Kenya
[5]World Vision International, Washington, DC, USA

**Correspondence to**
Dr Jessie Pinchoff;
jpinchoff@popcouncil.org

## ABSTRACT

**Objectives** COVID-19 may spread rapidly in densely populated urban informal settlements. Kenya swiftly implemented mitigation policies; we assess the economic, social and health-related harm disproportionately impacting women.

**Design** A prospective longitudinal cohort study with repeated mobile phone surveys in April, May and June 2020.

**Participants and setting** 2009 households across five informal settlements in Nairobi, sampled from two previously interviewed cohorts.

**Primary and secondary outcome measures** Outcomes include food insecurity, risk of household violence and forgoing necessary health services due to the pandemic. Gender-stratified linear probability regression models were constructed to determine the factors associated with these outcomes.

**Results** By May, more women than men reported adverse effects of COVID-19 mitigation policies on their lives. Women were 6 percentage points more likely to skip a meal versus men (coefficient: 0.055; 95% CI 0.016 to 0.094), and those who had completely lost their income were 15 percentage points more likely versus those employed (coefficient: 0.154; 95% CI 0.125 to 0.184) to skip a meal. Compared with men, women were 8 percentage points more likely to report increased risk of household violence (coefficient: 0.079; 95% CI 0.028 to 0.130) and 6 percentage points more likely to forgo necessary healthcare (coefficient: 0.056; 95% CI 0.037 to 0.076).

**Conclusions** The pandemic rapidly and disproportionately impacted the lives of women. As Kenya reopens, policymakers must deploy assistance to ensure women in urban informal settlements are able to return to work, and get healthcare and services they need to not lose progress on gender equity made to date.

## Strengths and limitations of this study

► This is one of the first longitudinal cohort studies on COVID-19 vulnerability and risks in informal settlements in a sub-Saharan African city.

► These findings highlight the disproportionate risks and impacts shouldered by women in the COVID-19 pandemic, and identify key characteristics associated with food insecurity, risk of household violence and forgoing necessary medical care by gender. By doing this study in partnership with government actors involved in the COVID-19 response in Kenya, these results will inform social support programmes in response to this pandemic to ensure the most vulnerable receive targeted assistance, with attention to the needs of women.

► There are two limitations: first, since participants of this study were sampled from existing cohorts from ongoing adolescent studies, the study population may not be fully representative of the setting population. Households were only eligible if an adolescent resided in the home; therefore, some households are not represented. Second, the survey was conducted among those we were able to reach by mobile phone, and responses are self-reported, which can lead to reporting and selection bias.

## INTRODUCTION

COVID-19, a highly transmissible respiratory disease caused by the novel SARS-CoV-2, was officially declared a pandemic on 11 March 2020 by the WHO.[1] Despite the lower than expected transmission to date, there is potential for a high number of cases and COVID-19 deaths in sub-Saharan Africa if containment and mitigation efforts fail.[2] Sub-Saharan Africa has rapidly urbanised; 47% of urban residents in the region live in informal settlements that are ill equipped to handle a disease outbreak.[3 4] These areas face increased risk

for rapid viral spread due to high population density, inadequate housing and sanitation facilities and intense levels of social mixing.[5] Coupled with the higher risks of disease transmission, residents of informal settlements face the concurrent shock of devastating impacts from physical distancing policies and lockdowns, including closing of businesses and schools and ban on large gatherings. Mitigation policies may slow the transmission of the virus, but they come with a heavy social and economic toll on poor urban populations, potentially higher for women.[6] There has been extensive theorising regarding how COVID-19 will impact African populations to date, but little systematic research. This study is one of the first prospective cohort studies conducted during COVID-19 to understand the experiences of those residing in urban informal settlements, including a broad set of indicators and measures across sectors and disciplines to holistically understand the interconnected short-term and long-term effects of the pandemic.

In Kenya, the first case of COVID-19 was detected on 13 March 2020 and resulted in the Kenyan government directing the immediate closure of schools and restaurants/bars and the prohibition of large gatherings. Two weeks later, on 26 March 2020, Kenya banned international flights.[7] On 6 April 2020, the Nairobi Metropolitan Area and three counties in coastal Kenya were contained, restricting movement into and out of these counties. Many businesses and stores closed as a result. Three months into the crisis, Kenya has confirmed 6673 cases and 149 deaths related to COVID-19 (as of 1 July 2020).[8] An estimated 60%–70% of Nairobi's more than 4 million residents reside in urban informal settlements.[9] From 30 to 31 March 2020, Population Council's COVID-19 mobile phone-based survey among 2009 informant settlement dwellers found 61% of respondents reported physical distancing measures would be challenging to follow, as it would risk their income.[10] The survey also found about 1 in 5 were worried about food shortages (22%) and about 1 in 3 were worried about job or income loss (34%).[10] Lack of income may be a significant challenge if prices for food and other critical needs go up, as news outlets are reporting. Recent reports express concerns that COVID-19 and Kenya's mitigation policies may lead to severe setbacks in access to healthcare, as well as reverse progress to date in nutrition, immunisation, other diseases and gender equity.[11 12]

Urban poor households are highly vulnerable during crises, and when faced with severe external shocks, are less able to cope with the health and financial impacts, as seen in previous case studies.[13] Implementing physical distancing and personal hygiene measures recommended by the WHO[14] to curb the spread of the outbreak will be challenging if not impossible in urban informal settlements.[15] Those who reside in informal settlements are less likely to have access to potable water or a latrine in their home; in the Kibera informal settlement in Nairobi, residents have one latrine for 50–150 people.[16] While there is fear regarding the spread of COVID-19,

the potential harm caused by income loss, food insecurity and forgoing health services may be just as severe, particularly for women.[17] Women are less likely to be employed in informal settlements, and the employment they do have is often tenuous and informal even before COVID-19. One study found only 22% of adult men and 3.6% of adult women report salaried employment in Nairobi informal settlements.[18] Women also disproportionately take on unpaid care burdens of children and the elderly, potentially forcing them to exit the workforce. This may leave women and female-headed households particularly vulnerable or dependent on male partners. Experts have yet to determine the severity of economic and social impacts likely to result from disease prevention and containment methods. A holistic approach, meaning one that takes into account the simultaneous impacts on income, access to health services and experience of violence, and the linkages between them, is needed to understand these inter-related secondary effects.

During and after emergencies, conflicts or epidemics, women often face extreme challenges due to the gender inequality and discrimination that existed before and is exacerbated due to sudden shifts in gender roles and relations.[19] Studies have found that women are more likely than men to face increased insecurity, restricted mobility and other major challenges.[19] During the Ebola outbreak in Sierra Leone, women disproportionately lost their jobs compared with men and the related economic disruptions impacted girls' education with more girls being pressured to drop out over boys due to economic constraints.[20] Women also tend to take on more unpaid household labour such as cooking, cleaning and childcare during times of crises including COVID-19.[21] With physical distancing, there are concerns that gender-based violence (GBV) may increase. Distancing may lead to social isolation and reduce the safety of victims. Stress and coping mechanisms such as increased alcohol consumption may also lead to more instances of violence.[22] In China, reports of domestic violence tripled during lockdown,[23] while in South Africa, reports have increased with 87 000 reports of domestic violence recorded in the first week of lockdown alone, despite the ban on alcohol.[24 25] At the same time, services to support survivors are being disrupted.[21] These trends threaten the progress made towards gender equality and GBV reduction efforts.

Access to healthcare is another critical dimension that may be exacerbated by the pandemic for women in particular. People may not be able to afford healthcare due to unemployment caused by the pandemic, facilities may restrict patient volume to minimise infection risk, and even if healthcare is available, people may avoid seeking care due to fear of infection at clinics. While some preliminary studies show men may be more likely to die of COVID-19,[26–29] women are adversely impacted in other ways. Mobility restrictions and the cost of healthcare fees may disproportionately limit the ability of women to seek healthcare, impacting their health but also the health of children in their care. A recent report suggests 30 million

children's lives may be at risk if secondary effects on health systems are similar for COVID-19 as they were for Ebola.[30] For example, shortly after Ebola there was a rise in measles cases, due to the drop in vaccinations caused by the crisis.[31] There is already the potential for this pattern in Kenya where already, compared with 2018 and 2019, under-5 outpatient department attendance and vaccination rates are significantly down.[29] Diverting resources to emerging threats can lead to neglecting other infectious diseases resulting in new waves of disease outbreaks and many lives lost, or shifting priorities that may make it more difficult to access sexual and reproductive health services.[21] These lessons need to be considered when allocating resources for COVID-19 and instating containment policies.

The primary objective of this study is to assess the short-term economic, social and health effects of COVID-19 and related mitigation measures among a prospective, longitudinal cohort of households sampled from five of Nairobi's informal settlements, with a focus on disproportionate burden placed on women. This cohort study was conducted rapidly during the initial COVID-19 response in Kenya, developed with the aim of understanding the experiences of households and the dynamic effects of the pandemic as the situation evolves. We evaluated disparities by gender, as research from previous health and humanitarian crises suggests women are likely to be disproportionately impacted. Our findings will help develop and better target both short-term and long-term policy and interventions. It will also highlight the prevalence of gender inequity and how this may impact the trajectories of women's lives.

## METHODS

The Population Council, in collaboration with the Kenyan Ministry of Health, has conducted mobile phone surveys across five urban informal settlements in Nairobi (Kibera, Mathare, Kariobangi, Dandora and Huruma). The study methods have been described elsewhere.[10] The study participants were randomly sampled from two existing longitudinal cohort studies, the Adolescent Girls Initiative-Kenya (AGI-K) and NISITU. Prior to COVID-19, the AGI-K urban cohort in Kibera and Huruma repeatedly sampled 2565 randomly selected households that had at least one adolescent resident. This cohort was part of a four-arm randomised controlled trial testing the impact of adolescent girl programmes. This included a baseline data collection in 2015, a second round of data collection in 2017 and a third round in 2019.[32] The NISITU cohort consisted of 4519 households randomly sampled from households in Kariobangi, Dandora and Mathare informal settlements. NISITU is a quasiexperimental study evaluating the effects of a gender transformative programme for girls, boys and young men. The NISITU baseline was conducted in early 2018 and the second wave of data collection in late 2019. NISITU and AGI-K mainly served as a sampling frame for the COVID-19 monthly surveys. For both cohorts, an initial household listing was conducted in the study sites to create a sampling frame of households with an eligible adolescent for the study. The last round of data collection for each was recently completed (September 2019 for AGI-K and in January 2020 for NISITU), therefore phone numbers for each head of household were up to date. All households were eligible for inclusion as long as an adult was reached on the phone. We randomly sampled from all 7500 households with available phone numbers (if multiple numbers were available, one was randomly selected), stratified by informal settlement. For the first round, we estimated a minimum of 400 participants per site was required at baseline (±5% CI calculation from a conservative 50% prevalence estimate for COVID-19-related knowledge). As the sample was randomly drawn from the pool of phone numbers, and there was no randomisation of intervention at this stage, no design effect of the study was considered. The COVID-19 cohort consists of about 2 009 adult household members interviewed on 30–31 March (round 1). We reinterviewed 1768 of these on 13–14 April (round 2) and 1750 on 10–11 May 2020 (round 3). Due to ongoing physical distancing policies in Kenya, all surveys were conducted on the phone to protect both participants and surveyors from potential COVID-19 spread. Each survey lasted about 30 min, with some questions asked across all three surveys and some unique questions in each of the three surveys.

### Survey instruments

The COVID-19 questionnaires used adapted standardised questions wherever possible, such as questions from the Demographic and Health Survey and the WHO/UNICEF Joint Monitoring Program on water and sanitation. A total of 77 Kenyan surveyors were trained on conducting mobile phone surveys, and before each survey round a training session was held to review the new questions. The first-round questionnaire included questions on basic demographics, awareness of COVID-19, knowledge, perceived risk, preventive behaviours being implemented, channels of information and trustworthiness of each source. In round 2, questions regarding social and economic effects on households were added, for example, loss of employment, skipping meals, household costs and GBV or tensions experienced in the household. In round 3, more detailed questions were added, such as how often participants are skipping meals. Each surveyor completed 10–15 surveys per day, and all phone numbers were tried up to three times if not reached on the first attempt. Questions were framed to reduce bias as much as possible. Data were collected using Open Data Kit and exported to Stata V.15 for analysis.

### Participant involvement

It was not possible to involve participants in the study design or interpretation of results due to the rapid response required around COVID-19, the inability to

engage face to face or hold events and that no funds were allocated for this.

Questionnaires and reports are publicly available, with the full deidentified data set available on request.

## Data analysis

Data from AGI-K and NISITU 2019 surveys were merged with rounds 1, 2 and 3 of the COVID-19 survey for additional information on household characteristics. The overall analysis here focuses on the three rounds of the COVID-19 survey; we incorporated basic information from the pre-COVID-19 surveys (such as wealth quintile, household location, household composition), but otherwise comparisons over time refer to the COVID-19 surveys since the same respondent was interviewed. Key economic, social and health variables were tabulated by survey round and gender. The survey question asked participants to self-identify their sex as male or female; throughout this paper we will refer to respondents as men and women to illustrate that we explore how the pandemic impacts gender (the socially constructed characteristics of men and women) not biological sex.

Key economic variables included reporting increased food prices, loss of income/employment and skipping a meal due to the pandemic. Key social variables included seeing family and friends less due to COVID-19, or taking on more chores such as cooking, cleaning and childcare. Participants were asked if they experienced more arguing, tension, household violence, or fear their partner would harm them due to COVID-19 (four binary questions). Key health variables focused on whether participants reported that in the last 2 weeks, they had required a health service (eg, for malaria, acute illness, immunisation or others) but did not seek healthcare for themselves or their children. For analysis, participants who said yes to one or more of these questions were considered at risk of household violence. Very few questions had missing responses.

Linear probability regression models were constructed to explore: (1) factors associated with skipping meals due to COVID-19, (2) factors associated with risk of household violence, and (3) factors associated with forgoing any health services due to COVID-19. Bivariate (unadjusted) models were constructed first followed by a fully adjusted model. Fully adjusted models adjust for gender, age, wealth quintile and informal settlement where the participant resides. In the models for skipping meals and forgoing health services, data were available for both April and May; in these models, clustering by participant was specified to allow the errors for the same participant to be correlated across each round over time. For the household violence risk model, these data were only collected from one time point (May). Models were then run stratified by gender. Linear probability regression models produce similar results to logistic regression, but allow for easier interpretation of the results in percentage points.

## Patient and public involvement

It was not possible to involve participants in the study design or interpretation of results due to the rapid response required around COVID-19, the inability to engage face to face or hold events, and that no funds were allocated for this. Questionnaires and reports are publicly available, with the full deidentified data set available upon request.

## RESULTS

### Characteristics of respondents

A total of 2009 individuals were surveyed in March, 1761 in April (88% of round 1) and 1745 in May (99% of round 2 participants; 87% of round 1 participants). Overall, participants were 63% women, with an average age of 36.3 (SD=±11.4) (table 1).

Most participants had completed secondary school (44%) and most were married (59%); however, there was variation by gender with men more likely to have achieved a higher educational level and more likely to be married. More than half of survey respondents reported they were the head of household, more so for men (83%) than women (41.3%). Between April and May 2020, significant changes in some social, health and economic effects were noted. Almost half of participants reported complete loss of income (43% in April vs 36% in May), experiencing more violence in the household (5% in April vs 3% in May) and increased reporting of skipping meals in the last week due to COVID-19 (74% vs 68%) (table 2). Despite some changes between round 2 and round 3, most differences by gender stayed the same, for example, the proportion of women skipping meals was higher than men in both survey rounds.

There was no change in the proportion reporting that they were not seeking health services they needed, this was 9% in both April and May, but overall higher for women compared with men. In round 3, questions also assessed measures of stigma related to COVID-19 infection, with significantly more women than men reporting that if infected people would gossip about them, and treat their family badly. Women were also less likely to say people would bring them food or medicine if needed (figure 1).

### Food security

Overall the majority of participants reported skipping a meal in the last week due to COVID-19, with some variation in the characteristics of those who skipped a meal. In fully adjusted models for April and May survey rounds, women were 6 percentage points more likely than men to report skipping meals in the past week due to COVID-19 (coefficient: 0.055; 95% CI 0.016 to 0.094) (table 3).

Those who reported a complete loss of income were 15 percentage points more likely to skip a meal (coefficient: 0.154; 95% CI 0.125 to 0.184). Compared with married couples, participants who were single were less likely to report skipping meals (coefficient: −0.059; 95% CI −0.118 to −0.000) and those who were divorced or widowed more likely to report skipping meals (coefficient 0.077;

**Table 1** Demographic characteristics of adult study sample selected in March 2020

| Demographics | Women n (%) | Men n (%) | Total (%) |
|---|---|---|---|
| n | 1258 (62.8) | 747 (37.2) | 2009 (100) |
| Age (years) | | | |
| 18–24 | 252 (20.0) | 181 (24.3) | 433 (21.6) |
| 25–34 | 285 (22.7) | 109 (14.6) | 394 (19.6) |
| 35–44 | 457 (36.3) | 243 (32.6) | 700 (34.8) |
| 45 and over | 264 (21.0) | 457 (36.3) | 477 (23.7) |
| Informal settlement | | | |
| Dandora | 242 (19.2) | 217 (29.1) | 459 (22.9) |
| Huruma | 208 (16.5) | 67 (9.0) | 275 (13.7) |
| Kariobangi | 223 (17.7) | 191 (25.6) | 414 (20.6) |
| Kibera | 322 (25.6) | 118 (15.8) | 440 (21.9) |
| Mathare | 263 (20.9) | 153 (20.5) | 416 (20.7) |
| Education | | | |
| No school | 59 (4.7) | 14 (1.9) | 73 (3.6) |
| Primary | 572 (45.5) | 221 (29.7) | 793 (39.6) |
| Secondary | 518 (41.2) | 360 (48.3) | 878 (43.8) |
| Higher education | 107 (8.5) | 150 (20.1) | 257 (12.8) |
| Marital status | | | |
| Married | 632 (50.4) | 538 (72.2) | 1170 (58.5) |
| Single | 313 (24.9) | 189 (25.4) | 502 (25.1) |
| Divorced/separated/widowed | 310 (24.7) | 18 (2.4) | 328 (16.4) |
| Respondent is head of household | 519 (41.3) | 619 (83.0) | 1138 (56.8) |
| Household size | | | |
| 1–2 people | 85 (6.8) | 186 (24.9) | 271 (13.5) |
| 3–4 people | 436 (34.7) | 188 (25.2) | 624 (31.1) |
| 5–6 people | 485 (38.6) | 254 (34.1) | 739 (36.9) |
| 7+ people | 252 (20.0) | 118 (15.8) | 370 (18.5) |

95% CI 0.031 to 0.122). Reportedly skipping meals due to COVID-19 also increased from April to May survey rounds; compared with April, participants in May were 5 percentage points more likely to report skipping a meal (coefficient: 0.049; 95% CI 0.023 to 0.074).

In models stratified by gender, both men and women were more likely to skip meals if they lost employment. Among female respondents, women who are divorced, widowed or separated were more likely to skip a meal than women who are married (coefficient: 0.080; 95% CI 0.032 to 0.128). Among men, having a larger household with more members was associated with an increased probability of skipping meals.

### Risk of household violence

In May, households reported more tension, arguing, violence, or fear their partner would harm them; combined, these indicate increased risk of household violence. In fully adjusted models, women were 8 percentage points more likely to report concerns regarding household tension and violence (coefficient: 0.079; 95% CI 0.028 to 0.130) (table 4).

Married and/or cohabiting couples were most likely to report this concern. Participants who had skipped a meal in the last 7 days due to COVID-19 were 16 percentage points more likely to report increased risk of household violence (coefficient: 0.164; 95% CI 0.111 to 0.218) and those who reported lost income were 7 percentage points more likely to report household tension and violence (coefficient: 0.066; 95% CI 0.019 to 0.114). In models stratified by gender, men who had completely lost their income due to COVID-19 were 14 percentage points more likely to report that they experienced increased tension in the household (coefficient: 0.140; 95% CI 0.063 to 0.218) while this was not significant among women.

### Forgoing needed health services

In fully adjusted models, women were 6 percentage points more likely to report not seeking health services they required (eg, for acute illness, for malaria treatment, for

**Table 2** Economic, social and health effects of mitigation measures in April (round 2) and May (round 3)

| | April (round 2) | | | May (round 3) | | | |
|---|---|---|---|---|---|---|---|
| | Women (%) | Men (%) | P value | Women (%) | Men (%) | P value | Round p value |
| **Economic effects** | | | | | | | |
| Increase in food prices | 870 (78.3) | 498 (76.6) | 0.41 | 927 (84.0) | 522 (81.3) | 0.14 | <0.001 |
| Complete loss of income | 422 (38.0) | 216 (33.2) | 0.05 | 515 (46.7) | 230 (35.8) | <0.001 | <0.001 |
| Skipped a meal this week due to COVID-19 | 775 (70.9) | 402 (62.9) | 0.001 | 838 (77.0) | 433 (68.0) | <0.001 | <0.001 |
| **Social effects** | | | | | | | |
| See family less | 607 (54.6) | 382 (58.8) | 0.09 | 621 (56.3) | 360 (56.1) | 0.93 | 0.946 |
| See my friends less | 968 (87.1) | 565 (86.9) | 0.9 | 1018 (92.3) | 565 (88.0) | 0.003 | 0.001 |
| *Risk of household violence* | | | | | | | |
| More tensions in the household | 434 (39.1) | 211 (32.5) | 0.01 | 400 (36.3) | 199 (31.0) | 0.03 | 0.166 |
| More violence experienced inside the house | 38 (3.5) | 20 (3.1) | 0.7 | 68 (6.2) | 18 (2.8) | 0.002 | 0.019 |
| More time arguing in the household | – | – | – | 307 (27.8) | 141 (22.0) | 0.007 | NA |
| More fear partner will harm you | – | – | – | 74 (6.7) | 29 (4.5) | 0.06 | NA |
| Combined into risk of household violence | – | – | – | 495 (44.9) | 248 (38.6) | 0.01 | NA |
| *Household chores* | | | | | | | |
| More housework (general) | 747 (67.2) | 330 (50.1) | <0.001 | – | – | – | NA |
| More time spent cooking | – | – | – | 539 (48.9) | 152 (23.7) | <0.001 | NA |
| More time spent cleaning | – | – | – | 670 (60.7) | 162 (25.2) | <0.001 | NA |
| More time spent taking care of children | – | – | – | 727 (65.9) | 230 (35.8) | <0.001 | NA |
| *Neighbourhood crime, violence* | | | | | | | |
| Increased crime in the neighbourhood | 426 (38.3) | 222 (34.2) | 0.08 | 557 (50.5) | 294 (45.8) | 0.06 | <0.001 |
| More violence experienced outside the house | 168 (15.1) | 92 (14.2) | 0.58 | 263 (23.8) | 146 (22.7) | 0.6 | <0.001 |
| **Health effects** | | | | | | | |
| Not purchasing sanitary pads (women only) | 405 (36.5) | – | NA | 451 (41.0) | – | NA | 0.030 |
| Not accessing needed healthcare/medicines | 124 (11.2) | 30 (4.6) | <0.001 | 117 (10.6) | 38 (5.9) | 0.001 | 1.000 |

NA, not applicable.

family planning) in the last 2 weeks (coefficient: 0.057; 95% CI 0.037 to 0.076) (table 5).

Those who had skipped a meal due to COVID-19 were 5 percentage points more likely to have forgone health services (coefficient 0.049; 95% CI 0.030 to 0.068) and those who had completely lost employment were also more likely to skip necessary health services (coefficient: 0.042; 95% CI 0.020 to 0.064). There was no significant

difference between April (round 2) and May (round 3) responses. Among those who skipped health services, over half said the reason was cost (52%). Participants who skipped health services were most likely to say the service they skipped was for any acute illness (27% in April, 21% in May), followed by seeking refills for any medications taken (19% in April, 20% in May).

### Receiving assistance

Between April and May survey rounds, the proportion of respondents receiving any assistance in the previous week tripled (7% in April to 21% in May); however, the proportion who had received assistance reporting that the assistance received covered their current needs decreased (47% in April to 38% in May) (figure 2). Most participants reported receiving soap or hand sanitiser, followed by food. When asked their single biggest need not being met, the main two items reported were food (94% in April, 86% in May) and cash (45% in April, 48% in May). Items were received from the government, non-governmental organisations (NGOs), good Samaritans/corporate sponsorship or religious institutions.

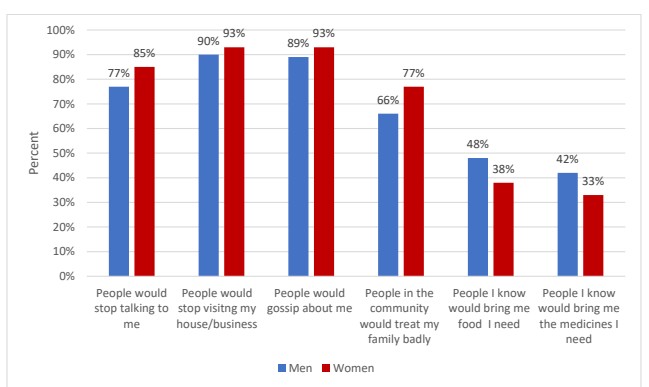

**Figure 1** Proportion of participants reporting stigma and social support related to COVID-19 by gender as measured in May (round 3).

**Table 3** Linear regression models of factors associated with skipping meals in the last week due to COVID-19

| Variables | Unadjusted models Coefficients (95% CI) | Adjusted model† Coefficients (95% CI) | Adjusted model: men only Coefficients (95% CI) | Adjusted model: women only Coefficients (95% CI) |
|---|---|---|---|---|
| Observations | | 3451 | 1274 | 2177 |
| Women (vs men) | 0.085 (0.049 to 0.122)*** | 0.055 (0.016 to 0.094)*** | – | – |
| **Age category (years)** | | | | |
| 18–24 | Ref | Ref | Ref | Ref |
| 25–34 | 0.147 (0.092 to 0.202)*** | 0.050 (−0.013 to 0.114) | −0.051 (−0.160 to 0.058) | 0.120 (0.040 to 0.199)*** |
| 35–44 | 0.150 (0.100 to 0.199)*** | 0.047 (−0.018 to 0.112) | −0.080 (−0.201 to 0.041) | 0.110 (0.030 to 0.190)*** |
| 45+ | 0.087 (0.032 to 0.141)*** | −0.002 (−0.071 to 0.067) | −0.107 (−0.232 to 0.018)* | 0.049 (−0.037 to 0.135) |
| **Marital status** | | | | |
| Married | Ref | Ref | Ref | Ref |
| Single | −0.109 (−0.153 to −0.065)*** | −0.059 (−0.118 to −0.000)** | −0.178 (−0.289 to −0.066)*** | −0.008 (−0.076 to 0.061) |
| Divorced/separated/widowed | 0.081 (0.039 to 0.123)*** | 0.077 (0.031 to 0.122)*** | 0.081 (−0.101 to 0.264) | 0.080 (0.032 to 0.128)*** |
| **Household size** | | | | |
| 1–2 people | Ref | Ref | Ref | Ref |
| 3–4 people | 0.048 (−0.012 to 0.108) | −0.008 (−0.069 to 0.053) | 0.003 (−0.086 to 0.091) | −0.043 (−0.133 to 0.047) |
| 5–6 people | 0.111 (0.054 to 0.168)*** | 0.049 (−0.011 to 0.110) | 0.070 (−0.018 to 0.159) | 0.018 (−0.072 to 0.107) |
| 7+ people | 0.112 (0.048 to 0.176)*** | 0.065 (−0.001 to 0.132)* | 0.104 (0.005 to 0.203)** | 0.032 (−0.064 to 0.128) |
| Complete loss of income due to COVID-19 | 0.172 (0.141 to 0.203)*** | 0.154 (0.125 to 0.184)*** | 0.200 (0.149 to 0.251)*** | 0.125 (0.089 to 0.162)*** |
| Survey round (2 vs 3) | 0.057 (0.027 to 0.088)*** | 0.049 (0.023 to 0.074)*** | 0.048 (0.004 to 0.092)** | 0.051 (0.020 to 0.083)*** |

*P<0.05; **p<0.01; ***p<0.001.
†Adjusted models also control for informal settlement where participant resides.

## DISCUSSION

Overall, women in informal settlements in Nairobi are disproportionately affected by mitigation policies implemented to reduce the spread of COVID-19. Compared with men, women are more likely to report they have skipped meals, report heightened risk of household violence and to report forgoing necessary healthcare services (for themselves or their children) due to the pandemic. Women are also more likely to report that if infected with COVID-19, they would experience stigma and have less social support. Between April and May, the proportion of respondents reporting that they are receiving government or NGO assistance increased, but the proportion reporting that this assistance is enough to meet their basic needs decreased. Our findings suggest that 3 months into the pandemic response, households in

**Table 4** Linear regression models of factors associated with reporting increased risk of household violence

| Variables | Unadjusted models Coefficients (95% CI) | Adjusted model† Coefficients (95% CI) | Men Coefficients (95% CI) | Women Coefficients (95% CI) |
|---|---|---|---|---|
| Observations | | 1775 | 636 | 1085 |
| Women (vs men) | 0.062 (0.015 to 0.110)** | 0.079 (0.028 to 0.130)*** | – | – |
| **Age (years)** | | | | |
| 18–24 | Ref | Ref | Ref | Ref |
| 25–34 | 0.121 (0.048 to 0.193)*** | 0.034 (−0.055 to 0.122) | −0.152 (−0.296 to −0.008)** | 0.131 (0.016 to 0.245)** |
| 35–44 | 0.081 (0.017 to 0.144)** | 0.015 (−0.073 to 0.104) | −0.038 (−0.191 to 0.114) | 0.044 (−0.069 to 0.156) |
| 45+ | 0.015 (−0.054 to 0.084) | −0.019 (−0.112 to 0.074) | −0.106 (−0.265 to 0.052) | 0.026 (−0.094 to 0.147) |
| **Marital status** | | | | |
| Married | Ref | Ref | Ref | Ref |
| Single | −0.089 (−0.145 to −0.034)*** | −0.073 (−0.152 to 0.006)* | −0.194 (−0.336 to −0.053)*** | 0.011 (−0.088 to 0.109) |
| Divorced/separated/widowed | −0.090 (−0.152 to −0.026)*** | −0.106 (−0.172 to −0.039)*** | −0.073 (−0.314 to 0.168) | −0.078 (−0.151 to −0.005)** |
| Has skipped a meal this week due to COVID-19 | 0.198 (0.147 to 0.250)*** | 0.164 (0.111 to 0.218)*** | 0.172 (0.090 to 0.254)*** | 0.140 (0.069 to 0.212)*** |
| Complete loss of income | 0.102 (0.055 to 0.148)*** | 0.066 (0.019 to 0.113)*** | 0.140 (0.063 to 0.218)*** | 0.035 (−0.025 to 0.095) |

*P<0.05; **p<0.01; ***p<0.001.
†Adjusted models also control for informal settlement where participant resides.

**Table 5** Linear regression models of factors associated with forgoing health services in the last 2 weeks

| Variables | Unadjusted models Coefficients (95% CI) | Adjusted model† Coefficients (95% CI) |
|---|---|---|
| Observations | | 3455 |
| Women (vs men) | 0.056 (0.037 to 0.076)*** | 0.057 (0.037 to 0.076)*** |
| Age (years) | | |
| 18–24 | Ref | Ref |
| 25–34 | 0.011 (−0.018 to 0.041) | −0.008 (−0.042 to 0.026) |
| 35–44 | −0.013 (−0.039 to 0.103) | −0.025 (−0.053 to 0.003)* |
| 45+ | −0.001 (−0.029 to 0.027) | −0.003 (−0.034 to 0.028) |
| Skipped a meal this week due to COVID-19 | 0.060 (0.040 to 0.081)*** | 0.049 (0.030 to 0.068)*** |
| Complete loss of income | 0.051 (0.032 to 0.071)*** | 0.042 (0.020 to 0.064)*** |
| Survey round (2 vs 3) | 0.001 (−0.017 to 0.020) | −0.004 (−0.022 to 0.014) |

*P<0.05; **p<0.01; ***p<0.001.
†Adjusted models also control for informal settlement where participant resides.

urban informal settlements in Nairobi are already experiencing significant economic, social and health-related harms. Women are disproportionately being impacted, and the effects of lost income, increased household chores and social isolation may compound to exacerbate and reinforce existing gender inequalities. Without intervention, the effects on health and well-being may be severe and set back progress made on gender inequality to date. As Kenya begins a phased reopening (from October 2020), it will be critical to support women to return to the workplace or to school and to facilitate improved gender dynamics as well as access to support services for GBV survivors.

### Economic impacts

Our study identified the serious threat of food insecurity 3 months into the pandemic response in Nairobi, with almost three-quarters of all study participants reporting they skipped a meal in the previous week due to COVID-19, higher for women than men. In crisis settings, women disproportionately bear the brunt of the effects of food insecurity with both short-term and long-term implications for health and well-being.[12 33 34] Due to pandemic mitigation policies and movement restrictions, county

government entities across Kenya have begun to close local, informal, outdoor markets where a majority of food is distributed.[35] These disruptions are likely to be causing supply shortages and price increases including in urban areas and reportedly making it difficult to access food, especially for families who cannot afford the higher food prices due to lost unemployment.[12 35] An African Union report projects that up to 20 million jobs could be lost in the region due to COVID-19.[36] Our study found that divorced women are at a higher risk of food insecurity compared with married and single women. This is perhaps due to the challenges a divorced woman faces as the single earner in the household while still also responsible for household chores including taking care of children, cleaning, laundry and cooking.[37 38] Further research should explore this connection.

### Social impacts

Our study found that overall households are experiencing increased risk of household violence. This may be related to forced time spent in the home, as well as widespread and sudden loss of employment leading to increased tension and limited access to GBV support programmes. During COVID-19 lockdowns in China, domestic violence reports tripled as people were forced to stay home.[23 39] During the Ebola outbreak in West Africa, GBV support initiatives were defunded due to prioritisation of infection control, while in South Africa, rumours about COVID-19 led to a drop in utilisation of GBV support programmes.[39 40] Multiple studies have found that intimate partner violence risk increases when employment is lost, including when women earn more than their partners or during socioeconomic hardship due to conflict.[41–43] This supports our finding that men surveyed reported more household tension, arguing and household violence risk if they had lost employment due to COVID-19. Our findings also indicate that increased food insecurity might be associated with increased risk of violence; studies in South Africa and Ethiopia prior to COVID-19 that found a significant link between

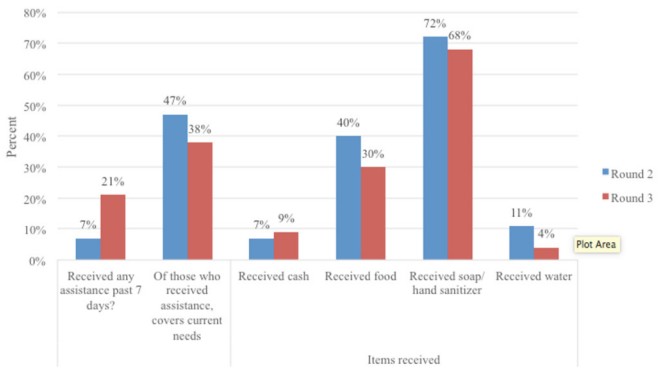

**Figure 2** Proportion of participants reporting assistance received and specific types of support received, comparing April (round 2) and May (round 3).

household food insecurity status and increased risk of violence support this connection.[44 45] To date, the Kenyan government has not coordinated any national response, such as food distributions, cash transfers or GBV support. The pandemic may exacerbate existing gender inequalities and roles, risking the fragile progress on these issues. As Kenya begins a phased reopening it is critical to understand how the pandemic is shaping gender dynamics and how to address disparities moving forward.

## Health impacts

In addition to food insecurity and household violence, there is concern that COVID-19 will lead to adverse secondary health effects if routine care including immunisation, nutrition, sexual and reproductive health and antenatal care is not used. More women than men reported that they had forgone health services due to the pandemic, potentially reflecting that women are responsible for seeking care for themselves and for their children. The main reason respondents gave for skipping a necessary health service was that they could not afford the service. People may not be able to afford necessary care due to unemployment and rising costs of food and other basic necessities. In 2013, the Kenyan Ministry of Health reported that 83% of Kenyans were not financially able to cover healthcare costs and 1.5 million Kenyans are driven into poverty each year because of this.[46] The economic insecurity that will result from COVID-19 will likely reverse any progress on this issue, causing potential outbreaks of other diseases and worsening health outcomes. Respondents also cited to a lesser degree concerns regarding stigma if infected and access challenges. Participants in the survey report clinics are reducing patient volume to reduce risk of infection. On the demand side, people may avoid seeking care if they are concerned they will get the disease, or that they will be forced to quarantine. According to the WHO, people who fear stigma related to COVID-19 infection may be driven to hide their illness, delay seeking healthcare and be less likely to adopt healthy behaviours.[47] Women in our study report forgoing critical health services for themselves and for their children, signalling the potential for secondary adverse health effects. The government and other stakeholders must prioritise ensuring affordable and accessible health services in a healthcare setting that is safe for providers and patients.

## Study limitations

This study has some limitations. First, we sampled from participants from AGI-K and NISITU cohorts, meaning that the COVID-19 cohort sample is not representative of all households but rather households in which an adolescent resides. This is because the initial cohorts (AGI-K and NISITU) were designed for different purposes, but allowed us a rich data set including mobile phone numbers to sample from. Therefore, for example, a household with only an elderly person or only early primary school-age children would not have been eligible for inclusion in

the COVID-19 survey. Second, all of the questions are self-reported, so there may be some bias in how participants respond to questions or recall bias. Relatedly, we interviewed one adult per household (whoever answered the phone), so their responses may not reflect all views or experiences of every household member. Other challenges relate to the phone-based nature of this survey, including some challenges verifying the respondent and had to rely on their responses. There also may have been some bias from who we could reach by phone, as some phone numbers never picked up. Another potential challenge with phone-based data collection is privacy; respondents may not have answered each question honestly if they were at home with others nearby. We tried to frame sensitive questions as yes/no responses so that respondents could answer honestly without others hearing the topic.

## Recommendations

To combat the negative economic, social and health-related impacts of COVID-19 mitigation measures, initiating government assistance in the form of cash transfers or food distributions may be critical, along with increasing safe and affordable access to needed health services to prevent secondary outbreaks. To date, there has not been a coordinated national response to the pandemic; any interventions are run by NGOs (primarily working with their constituents prior to COVID-19), by religious institutions and some regional leadership. Potential interventions to consider may include direct cash or asset transfers specifically to women, potentially using community-based women's groups as existing infrastructure for social and economic support so that the support is local and accessible. Direct cash transfer initiatives in other African countries show that efforts to target assistance for social protection and GBV directly to women and girls are more effective and worth exploring in this scenario.[48 49] Community-based women's groups have been shown to play an integral and effective role in addressing local health needs, including one study that found they reduced the burden of HIV in Zimbabwe; with proper precautions, these groups may be applied effectively in COVID-19 initiatives.[50–52] It is crucial to ensure that community health workers, mainly women themselves, are protected on the job and also able to provide gender-sensitive services, including antenatal care and family planning. Organisations such as United Nations Population Fund are fighting for these critical sexual and reproductive health services to be considered 'essential' so that resources and staffing are not reallocated due to COVID-19[53–55]; disruptions in these services have potentially devastating consequences for women and their families. Lastly, GBV hot lines or digital technologies (such as mobile phone-based apps) for violence prevention may increase accessibility to critical social services, but more research is required on their effectiveness and ability to reach the most vulnerable women.[56] Initiating these types of measures will also

help people sustain COVID-19 prevention measures over the longer term, for example, enabling them to safely stay home if another lockdown is implemented. These programmes can also ensure that women and their families are able to resume their pre-COVID-19 trajectory and that progress towards development goals to date is not lost or reversed. Attention to the needs of women in these settings is critical as they are disproportionately at risk. Additional research is necessary to understand how the pandemic may shape gender dynamics and power structures in the short term and long term, identifying opportunity for intervention and policy to continue to reduce inequalities. Our findings are likely to be broadly applicable to many other urban African contexts where there are large urban informal settlements and strict mitigation policies.

## CONCLUSION

Our findings suggest that 3 months into the COVID-19 pandemic and mitigation response, households in informal settlements are facing severe adverse social and economic effects, with women disproportionately impacted. Governments around the world are taking steps to balance the risks of COVID-19 transmission against the severe economic and social toll that will result from prolonged economic disruption. In the short-term, our findings highlight the urgent need to address rising unemployment, food insecurity and risk of GBV, all of which were more likely to be reported in this survey by women. Despite ongoing transmission of COVID-19, Kenya is entering a phased reopening. There is potential for additional disruptions if lockdowns or other mitigation policies return. Therefore, it will be critical to offer support programmes targeted to the most at-risk households and individuals as short-term stop gap measures, and in the longer term will be important to develop policies and programmes to support women as the pandemic has threatened gender inequity gains made to date. Future research and policy should address the long-term effects of the pandemic to understand how this event affected the life trajectories of urban women in informal settlements. Any larger coordinated policies or interventions must be gender sensitive to address the specific needs and situations of women.

**Acknowledgements** The authors acknowledge all of the work done by their field team to collect these surveys.

**Contributors** JP helped develop the study design, instrument, conducted the data analysis and led the drafting of the manuscript. KA led the study design, implementation, supported with developing the analysis plan and provided input on the manuscript. NR conducted the literature review, supported with data analysis and supported the drafting of the manuscript and tables. TA, BK and RO supported the study design, implementation and provided input on the manuscript. JBT supported with the study design, data management and analysis and input on the manuscript. DM, EM, FM and MN supported with study design, data collection, field team supervision and provided input on the manuscript. TDN conceived the study, helped develop the study design, analysis plan and provided input on the manuscript.

**Funding** UK Department for International Development through Innovations for Poverty Action Peace & Recovery COVID-19 rapid response grant (grant MIT0019-X15).

**Competing interests** None declared.

**Patient consent for publication** Not required.

**Ethics approval** We received expedited ethical approval via amendments to existing protocols to recontact respondents for the first two rapid surveys given the urgent need for information during this pandemic. The ongoing COVID-19 KAP survey cohort was approved by the Population Council IRB (p936) and AMREF ESRC (P803/2020). A research permit was issued by the National Commission for Science, Technology and Innovation (NACOSTI) in Kenya (P/20/5010). For AGI-K and NISITU, participants gave written informed consent. For the COVID-19 survey, verbal consent was given over the phone prior to administering the survey.

**Provenance and peer review** Not commissioned; externally peer reviewed.

**Data availability statement** Data are available upon reasonable request.

**ORCID iDs**
Jessie Pinchoff http://orcid.org/0000-0003-3155-595X
Karen Austrian http://orcid.org/0000-0001-5464-7908
Timothy Abuya http://orcid.org/0000-0001-8815-8299
James Benjamin Tidwell http://orcid.org/0000-0001-5868-6584

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
