## [Reviewer comments · BMJ Open]

ARTICLE DETAILS

TITLE (PROVISIONAL)	Gendered economic, social, and health effects of the COVID-19 pandemic and mitigation policies in Kenya: evidence from a prospective cohort survey in Nairobi informal settlements
AUTHORS	Pinchoff, Jessie; Austrian, Karen; Rajshekhar, Nandita; Abuya, Timothy; Kangwana, Beth; Ochako, Rhouné; Tidwell, James; Mwanga, Daniel; Muluve, Eva; Mbushi, Faith; Nzioki, Mercy; Ngo, Thoai

VERSION 1 – REVIEW

REVIEWER	Lenore Manderson University of the Witwatersrand
REVIEW RETURNED	14-Aug-2020

GENERAL COMMENTS	This article is very timely, and extremely important as evidence of the wider social and health impacts of COVID-19. I am very keen that it be published, but I also think it will benefit from revision, as indicated on the PDF (attached) and as set out below. (1) The discussion of methods makes it clear that the longitudinal nature of the study refers not to earlier cohorts (so comparing COVID with prior survey rounds) and so prior conditions within households, but that there were three waves of data collection using members of established cohorts. This is very clear in the abstract, but I think it would be helpful to strengthen clarity around these three rounds (March, April and May) and that these rounds generated the data under comparison. (2) The authors state that (to their knowledge) this is the first cohort study of populations impacted by COVID. This may well be the first to press, but given the recency of the pandemic, we don't know enough about who is doing what, and I don't think this point needs to be laboured. (3) I am not convinced of the value of the recurrent comparison with Ebola in the introductory sections and in the discussion, for many reasons. These include vast differences in infrastructure and living conditions between Sierra Leone (and Guinea and Liberia) and Kenya, the particular conditions of the informal settlements of Nairobi, and differences between the diseases (not least, marked differences in the mortality of infected cases). I don't mind a few comparisons but the obvious contrasts for COVID are with South Africa, and perhaps references to other dense areas of settlement. (4) More information about the settlements and their populations would be valuable – this is the context that allows for an holistic approach. The number of people living in precarious conditions, with poor infrastructure and services, in Nairobi is very high, and some more detail would be valuable given that many readers will not know this. It is also relevant to include some comment that many of the people living in the informal settlements in Nairobi work as day
---

	labourers, recycle, hustle and beg, and so the contraction of the economy is a direct result of limits to mobility; ie people did not have jobs to loose in the first place. (5) The authors refer to immunization, nutrition, sexual and reproductive health, and antenatal care, and they note the particular economic vulnerability of female headed households. But there are few details provided on compromises in health care. Are data available on health seeking for children? And given that the study populations are from households with adolescents, what do we know of this group? I have highlighted on the PDF a number of cases where the authors refer to children (including in single parent households) and it would be valuable to have more on this. On the other hand, the authors may be planning to write a separate article on dependents, which would be excellent. (6) I have suggested that the authors move recommendations regarding policies and interventions to the end of the article, and have marked where these appear earlier. (7) In this context, the authors refer to “access to mental health and gender-based violence support for the most vulnerable, particularly women and their children.” This is the only discussion of mental health, and given its low priority in Kenya (and everywhere) this either needs elaborating and contextualising, or should be deleted. Support for women and children subject to GBV (and extra-household sexual violence?) is worth recommending, but it is also relevant that Kenya (and other countries where it is prevalent, such as South Africa) struggle to find any intervention that works. (8) If the Kenyan government (or NGO community) stepped in during the study period to provide income support, grants, or food parcels (as in South Africa), then this is worth noting. It appears from the manuscript that this was not the case, (9) Some editing would tighten the text. I think phrases like “Some scholars have suggested that ...” are unnecessary, and tighter phrasing will sharpen the article and its messages. I also suggest that the authors address the use of present continuous and past continuous tense. I think simple past tense is clear and concise, and avoids speculating on what might still be happening. The reviewer provided a marked copy with additional comments. Please contact the publisher for full details.
--	--

REVIEWER	Rosemary Morgan Johns Hopkins Bloomberg School of Public Health
REVIEW RETURNED	01-Sep-2020

GENERAL COMMENTS	This is an important study which presents some very interesting data. The researchers were able to capitalize on an existing sample to collect data within informal settlements. The study presents more of a sex disaggregated analysis as opposed to a gender analysis. In order to be considered a gender analysis I would like to see more discussion about gender power relations and how these can lead to inequities between and among men and women. With some revisions I would recommend this article for publication. Within the abstract there is very little discussion of the gendered effects and differences between males and females. The introduction is very well written. I would recommend a slight restructuring, keeping the general content related to urban poor households together, and ending on the gendered impacts.
--

Within the paper gender is often conflated with sex. The authors appear to be conducting a sex disaggregated analysis of their data set, stratifying data by sex. Please include a definition of gender within the introduction or methods section and discuss how you have approached gender analysis. This would help to clarify the approach that is being taken. Gender analysis is more complex than disaggregating data by sex. How was gender factored into the design of the data collection tools? Was a gender framework used? It is clear that the authors are exploring gendered effects, i.e. how gender relations may create inequities in access to resources such as food, and increase in domestic labour, however, what is not clear is the thinking that went into the data collection and analysis in relation to gender. Note that male/female denotes sex and women/men denotes gender.

I would like to see more information related to age, not just the mean but the different age categories that were included.

Were any intersectional analyses conducted? For example, the percentage male and females across age categories, educational groups, or marital status? The demographic data has been aggregated which is not conducive to a gender analysis. I would like to see all data disaggregated by sex.

On page 8 you state “households reported more tension, arguing, violence, or fear their partner would harm them; combined these indicate increased risk of household violence.” Please disaggregated this data by sex.

On page 10 you state: “women in informal settlements in Nairobi are disproportionately affected by mitigation policies implemented to reduce the spread of COVID-19”. However, as much of the data in the results section was aggregated this didn’t come across as clearly as it could have. I would also like to see additional disaggregation by age and other social stratifiers if possible.

The findings summarized in the discussion are extremely important – I would like to see the sex disaggregated data clearly laid out in the results section.

Adding headings within the discussion section would make it easier to navigate, particularly in relation to social, health and economic effects. I would like to see some discussion about why differences between males and females were seen. I would also like to see more discussion about the context of informal settlements within the conclusion.

The COVID-19 cohort sample was representative of households in which an adolescent resides, however, there was little to no discussion of age differences or effect on adolescents in the results or discussion. Why is the fact that an adolescent resided in the household relevant?

Within the study, a prospective, longitudinal cohort of households were sampled, however, the longitudinal nature of the findings were not always clear. It is unclear what the relevant of the previous round of data collection were beyond the fact that this study drew from the same sample. If similar questions were not asked in the previous questionnaires can the change over time be measured beyond self reporting? More clarity is needed in relation to how the findings from

	the different rounds related to one another. It seems that the sample drawn from was for convenience in that it already existed and this may be more of a cross sectional study. The relevance of sampling households with adolescents is also unclear. Why was this important? Again, it seems that the researchers used a readily available sample (which is fine), however, the sample was designed for different purposes. This needs to be stated as a limitation. I would like to see more discussion about how the issues that were brought up can be addressed. The evidence clearly shows that there are gender inequities, but what can and should be done about it?
--	--

REVIEWER	Eddie T. C. Lam, Ph.D. Cleveland State University, USA
REVIEW RETURNED	27-Sep-2020

GENERAL COMMENTS	Overall, the manuscript is well written and timely during this COVID-19 crisis. The following comments aim at improving the readability of the manuscript so that it is more approachable to the readers.  1. The title is “Gendered social, health, and economic effects . . .” (please use a comma to separate “health” and “economic”). The information in the entire manuscript should be presented in the same order as the title (or vice versa). For instance, the manuscript is difficult to follow since the order of these three areas are different in different sections (i.e., in random order). 2. Briefly explain what “linear probability regression models” can do in data analysis (p. 6). Will multiple regression be more meaningful to determine the predictors of those outcome variables? 3. SD should follow the “±” sign (p. 7). For example, (SD = ±11.4). 4. For Table 2, all the numbers in parentheses should have the “%” sign (p. 8). For example, 992 (56.2%). 5. Results: elaborate what “percentage points” (pp) mean. Is “pp” a standardized unit across all models? 6. Discussion: again, discuss what those different “pp” mean in each setting. Any suggestions regarding how to address “unemployment, food insecurity and to strengthen existing structures at the community”?
---

VERSION 1 – AUTHOR RESPONSE

Reviewer(s)' Comments to Author:

Reviewer: 1

Reviewer Name: Lenore Manderson

Institution and Country: University of the Witwatersrand

Please state any competing interests or state 'None declared': None declared

Please leave your comments for the authors below

This article is very timely, and extremely important as evidence of the wider social and health impacts of COVID-19. I am very keen that it be published, but I also think it will benefit from revision, as indicated on the PDF (attached) and as set out below.

Please also see additional comments in attached PDF

(1) The discussion of methods makes it clear that the longitudinal nature of the study refers not to earlier cohorts (so comparing COVID with prior survey rounds) and so prior conditions within

households, but that there were three waves of data collection using members of established cohorts. This is very clear in the abstract, but I think it would be helpful to strengthen clarity around these three rounds (March, April and May) and that these rounds generated the data under comparison.

Correct, we sampled from previous rounds to create this new cohort. We randomly sampled households from the list of NISITU and AGI-K households for which we had a phone number and were told we could re-contact them in the future. We sampled 400 or more per informal settlement reaching our sample size of 2,009 for round 1 of the COVID-19 survey. We have added in the methods the round of each data collection (round 1= March 30, 31st; round 2 = April 13-14th; round 3 = May 10-11th).

In response to a comment in the PDF, we are leaving our longer description of AGI-K and NISITU because for NISITU in particular there is not a publication that clearly lays out the sampling, so we will include it here.

(2) The authors state that (to their knowledge) this is the first cohort study of populations impacted by COVID. This may well be the first to press, but given the recency of the pandemic, we don't know enough about who is doing what, and I don't think this point needs to be laboured.

We agree and have rephrased to state in the last paragraph of the background that regardless of other work being done, this survey was initiated very rapidly in the field.

(3) I am not convinced of the value of the recurrent comparison with Ebola in the introductory sections and in the discussion, for many reasons. These include vast differences in infrastructure and living conditions between Sierra Leone (and Guinea and Liberia) and Kenya, the particular conditions of the informal settlements of Nairobi, and differences between the diseases (not least, marked differences in the mortality of infected cases). I don't mind a few comparisons but the obvious contrasts for COVID are with South Africa, and perhaps references to other dense areas of settlement.

We have cut some of the Ebola comparisons and edited to discuss other situations and other urban areas.

(4) More information about the settlements and their populations would be valuable – this is the context that allows for an holistic approach. The number of people living in precarious conditions, with poor infrastructure and services, in Nairobi is very high, and some more detail would be valuable given that many readers will not know this. It is also relevant to include some comment that many of the people living in the informal settlements in Nairobi work as day labourers, recycle, hustle and beg, and so the contraction of the economy is a direct result of limits to mobility; ie people did not have jobs to loose in the first place.

This is certainly useful to add more detail on, we have fleshed out our description of the gender and economic context of the urban informal settlements.

(5) The authors refer to immunization, nutrition, sexual and reproductive health, and antenatal care, and they note the particular economic vulnerability of female headed households. But there are few details provided on compromises in health care. Are data available on health seeking for children? And given that the study populations are from households with adolescents, what do we know of this group? I have highlighted on the PDF a number of cases where the authors refer to children

(including in single parent households) and it would be valuable to have more on this. On the other hand, the authors may be planning to write a separate article on dependents, which would be excellent.

We asked respondents if they had skipped health services in the last 2 weeks, then followed up for those who said they had forgone services by asking which services. These included some for adults (e.g., family planning) and some that could be for any family member (e.g., malaria) or specifically for children (e.g., immunization). We do have a separate article regarding adolescents, and did ask them a similar question, but here it is not possible to tease apart who the services were for, and they are only asked of the subset who skipped necessary health services so further analysis was not possible.

(6) I have suggested that the authors move recommendations regarding policies and interventions to the end of the article, and have marked where these appear earlier.

We have created a recommendations section at the end and moved the sentences indicated into this section.

(7) In this context, the authors refer to “access to mental health and gender-based violence support for the most vulnerable, particularly women and their children.” This is the only discussion of mental health, and given its low priority in Kenya (and everywhere) this either needs elaborating and contextualising, or should be deleted.

Support for women and children subject to GBV (and extra-household sexual violence?) is worth recommending, but it is also relevant that Kenya (and other countries where it is prevalent, such as South Africa) struggle to find any intervention that works.

We have cut mental health here, since this was not a focus for this paper and the reviewer is correct that this is a complex issue regarding more attention.

We have also clarified that to date no GBV support program has been implemented in Kenya specifically in response to COVID-19 but that generally some type of targeted response may be warranted.

(8) If the Kenyan government (or NGO community) stepped in during the study period to provide income support, grants, or food parcels (as in South Africa), then this is worth noting. It appears from the manuscript that this was not the case,

We have added a sentence on page 11 to highlight that while some participants did report that they received money or food from a variety of sources (Government, NGO's, religious institutions), that it was not sufficient to meet their basic needs. We have also added a sentence in the discussion to highlight that the Kenyan government has NOT to date implemented any coordinated, national response.

(9) Some editing would tighten the text. I think phrases like “Some scholars have suggested that ...” are unnecessary, and tighter phrasing will sharpen the article and its messages. I also suggest that the authors address the use of present continuous and past continuous tense. I think simple past tense is clear and concise, and avoids speculating on what might still be happening.

We have edited accordingly to clarify the tense and make the writing more concise.

Reviewer: 2

Reviewer Name: Rosemary Morgan

Institution and Country: Johns Hopkins Bloomberg School of Public Health
Please state any competing interests or state 'None declared': None declared

Please leave your comments for the authors below

This is an important study which presents some very interesting data. The researchers were able to capitalize on an existing sample to collect data within informal settlements. The study presents more of a sex disaggregated analysis as opposed to a gender analysis. In order to be considered a gender analysis I would like to see more discussion about gender power relations and how these can lead to inequities between and among men and women. With some revisions I would recommend this article for publication.

Within the abstract there is very little discussion of the gendered effects and differences between males and females.

We have revised the abstract to focus more of the gendered effects.

The introduction is very well written. I would recommend a slight restructuring, keeping the general content related to urban poor households together, and ending on the gendered impacts.

We have revised the introduction based on reviewer feedback, and now end the introduction with the gendered impacts.

Within the paper gender is often conflated with sex. The authors appear to be conducting a sex disaggregated analysis of their data set, stratifying data by sex. Please include a definition of gender within the introduction or methods section and discuss how you have approached gender analysis. This would help to clarify the approach that is being taken. Gender analysis is more complex than disaggregating data by sex. How was gender factored into the design of the data collection tools? Was a gender framework used? It is clear that the authors are exploring gendered effects, i.e. how gender relations may create inequities in access to resources such as food, and increase in domestic labour, however, what is not clear is the thinking that went into the data collection and analysis in relation to gender. Note that male/female denotes sex and women/men denotes gender. Thank you for this feedback. We have added a sentence to the methods to define gender and how we use it in this paper: "The survey question asked participants to self identify their sex as male or female; throughout this paper we will refer to respondents as men and women to illustrate that we explore how the pandemic impacts gender (the socially constructed characteristics of men and women) not biological sex." We have also switched our terminology to men/women for gender instead of male/female.

Our data collection was conducted rapidly and initially designed to understand household needs more generally. While no gender framework was used, based on previous research from epidemic contexts we hypothesized early on that there may be differential effects of COVID-19 by gender, and included questions related to some of these potential areas of variation.

I would like to see more information related to age, not just the mean but the different age categories that were included.

We have edited table 1 to include the categories of age and removed the mean.

Were any intersectional analyses conducted? For example, the percentage male and females across age categories, educational groups, or marital status? The demographic data has been aggregated which is not conducive to a gender analysis. I would like to see all data disaggregated by sex.

This is a helpful comment we have disaggregated table 1 and 2 by gender to see the differences, including across age, education and marital status.

On page 8 you state “households reported more tension, arguing, violence, or fear their partner would harm them; combined these indicate increased risk of household violence.” Please disaggregated this data by sex.

This is now presented in table 2 disaggregated by sex; we show the proportion of men and women reporting each component of the combined household violence variable.

On page 10 you state: “women in informal settlements in Nairobi are disproportionately affected by mitigation policies implemented to reduce the spread of COVID-19”. However, as much of the data in the results section was aggregated this didn’t come across as clearly as it could have. I would also like to see additional disaggregation by age and other social stratifiers if possible.

We now present table 1 and 2 by gender, as suggested, to present the gender disaggregated sociodemographic information of respondents.

In the results section, when presenting our models, we first highlight the aggregated model results to show the difference by gender, then present separate models for men and women, to highlight effects that are significant within each gender group.

The findings summarized in the discussion are extremely important – I would like to see the sex disaggregated data clearly laid out in the results section.

As highlighted above, we have tried to more clearly layout the results by gender. Table 1 and 2 are now presented men/women/total, and the models are run for the whole sample and then stratified by gender.

Adding headings within the discussion section would make it easier to navigate, particularly in relation to social, health and economic effects. I would like to see some discussion about why differences between males and females were seen. I would also like to see more discussion about the context of informal settlements within the conclusion.

We have added headings to make the discussion easier to follow, and have added more details about gender differences reported and also the context of informal settlements throughout.

The COVID-19 cohort sample was representative of households in which an adolescent resides, however, there was little to no discussion of age differences or effect on adolescents in the results or discussion. Why is the fact that an adolescent resided in the household relevant?

We highlight with an example after this sentence in the limitations to show that households in our survey all have an adolescent, per the eligibility criteria for AGIK/NISITU. Households with an adolescent are more likely to be representative of a certain type and age family – for example, a household with only elderly people, or family with only school-aged children, would not be eligible. We just want to ensure it is clear that our sample is not representative of ALL households in informal settlements.

Within the study, a prospective, longitudinal cohort of households were sampled, however, the longitudinal nature of the findings were not always clear. It is unclear what the relevant of the previous round of data collection were beyond the fact that this study drew from the same sample. If similar questions were not asked in the previous questionnaires can the change over time be

measured beyond self reporting? More clarity is needed in relation to how the findings from the different rounds related to one another. It seems that the sample drawn from was for convenience in that it already existed and this may be more of a cross sectional study. The relevance of sampling households with adolescents is also unclear. Why was this important? Again, it seems that the researchers used a readily available sample (which is fine), however, the sample was designed for different purposes. This needs to be stated as a limitation.

- 1) Regarding the longitudinal nature of the dataset, we have added further clarification in the methods, specifically under data analysis. AGI-K and NISITU datasets mainly served as sampling frames, and longitudinal refers to changes between COVID-19 surveys not pre/post COVID-19.
- 2) Regarding the sampling, we have added a sentence to the limitations to clarify the sample was designed for different purposes but allowed us a sample to pull from including mobile phone numbers.

I would like to see more discussion about how the issues that were brought up can be addressed. The evidence clearly shows that there are gender inequities, but what can and should be done about it? we have added some more detail to the discussion and a recommendations paragraph to highlight what we suggest can be done to address these issues.

Reviewer: 3

Reviewer Name: Eddie T. C. Lam

Institution and Country: Cleveland State University, USA

Please state any competing interests or state 'None declared': None

Please leave your comments for the authors below

Overall, the manuscript is well written and timely during this COVID-19 crisis. The following comments aim at improving the readability of the manuscript so that it is more approachable to the readers.

1. The title is "Gendered social, health, and economic effects . . ." (please use a comma to separate "health" and "economic"). The information in the entire manuscript should be presented in the same order as the title (or vice versa). For instance, the manuscript is difficult to follow since the order of these three areas are different in different sections (i.e., in random order).

This is helpful for clarity – we have reorganized throughout and updated the title so that the order is economic, social, and health effects throughout.

2. Briefly explain what "linear probability regression models" can do in data analysis (p. 6). Will multiple regression be more meaningful to determine the predictors of those outcome variables?

We have added some detail, linear probability regression models are for binary outcome variables such as ours, and provide ease of interpretation (percentage point changes) and are similar to logistic regression models.

3. SD should follow the "±" sign (p. 7). For example, (SD = ±11.4).

We have revised accordingly.

4. For Table 2, all the numbers in parentheses should have the "%" sign (p. 8). For example, 992 (56.2%).

We have revised accordingly.

5. Results: elaborate what “percentage points” (pp) mean. Is “pp” a standardized unit across all models?

We have added this in the results to clarify we refer to percentage points because we use linear probability models that allow for us to show the percentage increase in units for each outcome.

6. Discussion: again, discuss what those different “pp” mean in each setting. Any suggestions regarding how to address “unemployment, food insecurity and to strengthen existing structures at the community”?

We have added some clarification that we are reporting the percentage point differences in the outcomes between groups. E.g., in a fully adjusted model, women were 6 percentage points more likely to report skipping a meal in the last week as compared to men.

We have also added some comments in the discussion in the recommendations section regarding how to address these challenges.

VERSION 2 – REVIEW

REVIEWER	Rosemary Morgan Johns Hopkins Bloomberg School of Public Health
REVIEW RETURNED	23-Dec-2020

GENERAL COMMENTS	This is an excellent and well written paper which addresses and important topic. The findings are robust and highlight important gender inequities related to COVID-19. I have recommendations to strengthen the paper as it is already very strong.
--

REVIEWER	Eddie T. C. Lam, Ph.D. Cleveland State University, USA
REVIEW RETURNED	25-Nov-2020

GENERAL COMMENTS	The authors have made significant improvement for this revision. Though the authors indicated that they had added some comments in the Discussion and Recommendations, I still don't see where and how they addressed “unemployment, food insecurity, and to strengthen existing structures at the community.” Besides, change all the "NGO's" in the manuscript to "NGOs" since the apostrophe (i.e., procession) should not be used for plurals.
--